# Scale-Aware Contrastive Reverse Distillation for Unsupervised Medical Anomaly Detection

**Chunlei Li** [*]
MedAI Technology (Wuxi) Co. Ltd.
chunlei.li@medimagingai.com

**Yilei Shi** [*]
MedAI Technology (Wuxi) Co. Ltd.
yilei.shi@medimagingai.com

**Jingliang Hu**
MedAI Technology (Wuxi) Co. Ltd.
jingliang.hu@medimagingai.com

**Xiao Xiang Zhu**
Technical University of Munich
xiaoxiang.zhu@tum.de

**Lichao Mou** [†]
MedAI Technology (Wuxi) Co. Ltd.
lichao.mou@medimagingai.com

## Abstract

Unsupervised anomaly detection using deep learning has garnered significant research attention due to its broad applicability, particularly in medical imaging where labeled anomalous data are scarce. While earlier approaches leverage generative models like autoencoders and generative adversarial networks (GANs), they often fall short due to overgeneralization. Recent methods explore various strategies, including memory banks, normalizing flows, self-supervised learning, and knowledge distillation, to enhance discrimination. Among these, knowledge distillation, particularly reverse distillation, has shown promise. Following this paradigm, we propose a novel scale-aware contrastive reverse distillation model that addresses two key limitations of existing reverse distillation methods: insufficient feature discriminability and inability to handle anomaly scale variations. Specifically, we introduce a contrastive student-teacher learning approach to derive more discriminative representations by generating and exploring out-of-normal distributions. Further, we design a scale adaptation mechanism to softly weight contrastive distillation losses at different scales to account for the scale variation issue. Extensive experiments on benchmark datasets demonstrate state-of-the-art performance, validating the efficacy of the proposed method. Code is available at url https://github.com/MedAITech/SCRD4AD.

## 1 Introduction

The automatic detection of anomalies in medical images is a crucial yet challenging task. Finding abnormalities early through screening enables timely intervention and improves patient outcomes (Cai et al., 2023). However, designing robust algorithms for this task is difficult due to the variability in anomalous anatomy across patients. Moreover, obtaining annotated data with verified anomalous samples is often prohibitively expensive and time-consuming (Cai et al., 2023; Schlegl et al., 2019). Consequently, there is a pressing need for unsupervised anomaly detection methods capable of recognizing anomalies without relying on labeled abnormal training data.

Early efforts utilize generative models such as autoencoders and generative adversarial networks (GANs) (Schlegl et al., 2019; Jiang et al., 2019; Han et al., 2021a; Shvetsova et al., 2021). These models are trained to learn feature representations exclusively from normal images. Abnormalities

---

[*] Equal contribution.
[†] Corresponding author.

can then be detected by determining if test images lie outside the manifold of the learned representations, or by comparing original and generated images in pixel space. The underlying hypothesis is that a generative model trained on normal samples can accurately reconstruct anomaly-free regions well but struggles with anomalous ones. Nevertheless, this does not always hold, and generative models often overgeneralize, that is, they tend to generalize too well, thereby risking the reconstruction of abnormal regions (Gong et al., 2019).

To address this issue, some approaches introduce memory banks that store representative normal patterns to help control model generalization (Gong et al., 2019). At test time, images are directly compared to the memory banks to identify anomalies. In addition, several studies use normalizing flows (Rudolph et al., 2021; Yu et al., 2021; Gudovskiy et al., 2022). A normalizing flow models a target distribution as an invertible transformation of a base distribution (e.g., Gaussian) in latent space. In the context of anomaly detection, the flow is trained to maximize the likelihood of normal patterns, and the likelihood cannot be simultaneously increased for all images. By doing so, it assigns positive density only to normal samples and, in particular, does not generalize to anomalous ones. Concurrently, self-supervised learning (Jing & Tian, 2021) has catalyzed the development of unsupervised anomaly detection algorithms (Li et al., 2021; Sohn et al., 2021; Schlüter et al., 2022). Existing self-supervised learning-based methods typically follow one of two paradigms: one-stage or two-stage approaches. In the one-stage approach, a model is trained to detect artificially synthesized anomalies and then directly applied to detect real abnormalities. The two-stage approach, on the other hand, first learns self-supervised representations through a pretext task on normal data and subsequently constructs a one-class classifier based on the learned representations. However, these methods still exhibit limited discriminative capabilities in real-world medical anomaly detection and incur heavy computational overheads.

Knowledge distillation from pre-trained models presents another promising approach for unsupervised anomaly detection (Salehi et al., 2021). This methodology typically employs a teacher-student paradigm, where the teacher is an encoder network pre-trained on a large-scale dataset (e.g., ImageNet), and the student network has a similar or identical architecture. The key insight is that the student is exposed only to anomaly-free images during knowledge distillation, leading to discrepancies between features of the teacher and student networks when encountering anomalies during inference. Knowledge distillation-based approaches reconstruct features of pre-trained encoders rather than raw pixels, as features provide more informative representations and yield superior results. To further enhance the discriminative capability of the teacher-student framework, various strategies are explored. For example, Bergmann et al. (2020) ensemble several student networks trained on normal images at different scales, while Salehi et al. (2021) and Wang et al. (2021) leverage multi-level feature alignment. Deng & Li (2022) propose an interesting reverse distillation model that adopts a heterogeneous teacher-student framework, comprising a teacher encoder and a student decoder. This method distills knowledge from the pre-trained teacher network into the student network in a reverse direction, achieving better performance.

In this paper, we approach the problem of unsupervised anomaly detection through the lens of reverse distillation. We identify two key limitations of the reverse distillation model. First, knowledge distillation alone is insufficient to provide discriminative representations to the student network. Second, we observe that anomalies vary in size, posing a challenge to scale-equalizing knowledge distillation models. To address these issues, we propose a scale-aware contrastive reverse distillation model. Our contributions are threefold:

- We propose a contrastive student-teacher learning method within the reverse distillation paradigm, aimed at deriving discriminative representations by generating and exploring out-of-normal data distributions.

- To address the scale variation issue, we propose a scale adaptation mechanism that learns to softly weight contrastive representation distillation at each scale.

- We evaluate the proposed approach on three datasets from different imaging modalities: X-ray, MRI, and dermoscopy. Experimental results demonstrate state-of-the-art performance across all datasets.

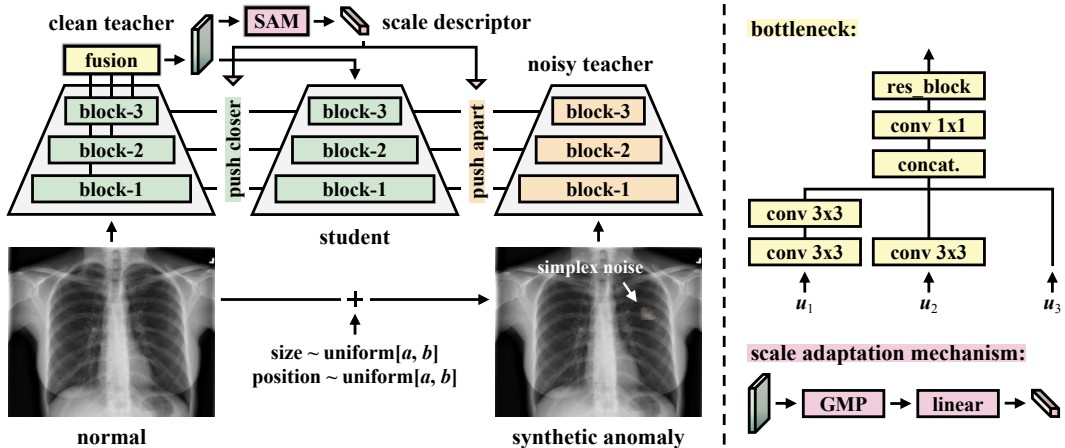

Figure 1: Illustration of the proposed framework during training. It comprises two distinct encoding pathways: 1) a "clean" teacher encoder followed by a bottleneck, a scale adaptation mechanism, and a student decoder, and 2) a "noisy" teacher encoder. The two teacher encoders share weights but process different inputs: the clean teacher receives normal data, whereas the noisy teacher processes synthesized anomalies. We employ contrastive reverse distillation by pushing the student's reconstructed features closer to feature representations from the clean teacher and farther from those of the noisy teacher. The scale adaptation module generates input-specific scale weights used in this process.

## 2 METHODOLOGY

### 2.1 PRELIMINARY

First, we revisit the original reverse distillation model for anomaly detection as proposed by Deng & Li (2022). It consists of three components: 1) a fixed pre-trained encoder serving as the teacher to extract feature maps from an input image; 2) a bottleneck fusing multi-scale features from the encoder into a joint representation; and 3) a decoder acting as the student to reconstruct feature maps at each scale from the joint representation.

Let $\text{sim}(\boldsymbol{a}, \boldsymbol{b}) = \boldsymbol{a}^{\mathrm{T}}\boldsymbol{b} / \|\boldsymbol{a}\| \|\boldsymbol{b}\|$ denote the dot product between $\ell_2$ normalized $\boldsymbol{a}$ and $\boldsymbol{b}$ (i.e., cosine similarity), and $\boldsymbol{u}_k$ and $\boldsymbol{v}_k$ be feature maps from the $k$-th layer of the teacher encoder and student decoder, respectively. For knowledge transfer in the model, the following loss is used:

$$\mathcal{L} = \sum_k \frac{1}{C_k} \sum_h \sum_w 1 - \text{sim}(\boldsymbol{u}_{k,h,w}, \boldsymbol{v}_{k,h,w}), \tag{1}$$

where $C_k$ is a normalization constant, and $(h, w)$ enumerates all integral spatial locations in feature maps $\boldsymbol{u}_k$ and $\boldsymbol{v}_k$. After training the teacher encoder and student decoder on normal data only, they align to represent normal patterns in a similar way. For normal test samples that conform to patterns observed during training, the student's representations closely match the teacher's, leading to low vector-wise cosine similarity losses. Conversely, for anomalous samples, the student decoder fails to properly reconstruct the teacher's feature maps due to having learned only normal patterns.

### 2.2 CONTRASTIVE REVERSE DISTILLATION

#### 2.2.1 NETWORK ARCHITECTURE AND LOSS

One limitation of the approach proposed by Deng & Li (2022) is that the teacher only sees normal data, i.e., in-distribution examples, during training and does not attempt to explore potential out-of-distribution representations. This may result in a lack of discrimination for anomalies in real-world scenarios. To address this, we propose a novel paradigm called contrastive reverse distillation for anomaly detection (see Figure 1). It introduces a second teacher encoder, termed "noisy" teacher,

which can synthesize plausible out-of-normal representations to serve as anomalous instances. The resulting model contains two distinct encoding pathways: one is a "clean" teacher encoder followed by a bottleneck and a student decoder as described in Section 2.1, and the other is the noisy teacher encoder. The two teacher encoders share weights but take different images as input—the clean teacher sees normal data while the noisy teacher sees synthesized anomalies.

Formally, let $x$ represent a normal training image, $x' = g(x)$ be a synthesized abnormal image, and $\phi$ refer to the output of the bottleneck, i.e., a multi-scale representation of $x$. We denote feature maps from the $k$-th layer of the clean teacher encoder, noisy teacher encoder, and student decoder as $u_k = f(x)$, $z_k = f(x')$, and $v_k$, respectively, where $f(\cdot)$ indicates the encoders. In our model, we aim to push $u_k$ and $v_k$ closer while pushing $z_k$ and $v_k$ apart. To this end, we design a novel contrastive reverse distillation loss:

$$\mathcal{L} = \sum_k \frac{1 - \text{sim}(u_k, v_k)}{1 - \text{sim}(z_k, v_k) + \epsilon} \, , \tag{2}$$

where $\epsilon$ is a small value added to the denominator to avoid division-by-zero errors in practice. It is worth noting that unlike Eq. (1), here $u_k$ and $v_k$ are reshaped into 1D representations prior to calculating the cosine similarity between them, and the same reshaping operation is applied to $z_k$ as well. This makes results more stable.

### 2.2.2 IMAGE SYNTHESIS FOR NOISY TEACHER

Medical images typically exhibit a power law distribution of frequencies, with lower frequency components dominating the image content (Wyatt et al., 2022). Based on the assumption that both normal and abnormal images adhere to this power law, we are motivated to synthesize abnormal images by applying noise with a similar distribution. While such noise can be generated through Gaussian random fields or engineered covariance matrices, we opt for a simpler approach utilizing simplex noise (Perlin, 2002), following methods of Tien et al. (2023) and Wyatt et al. (2022). Simplex noise enables precise control over the frequency distribution of images. In contrast to Gaussian noise, simplex noise produces smooth, structured randomness, making it well-suited for our task. We compare the performance of these two noise types in our model (cf. Section 3.3.3).

In our approach, for each training image $x$, we first sample noise size and position from uniform distributions. Then, simplex noise (Perlin, 2002) is generated with six octaves and a persistence of $\gamma = 0.6$. Finally, the generated noise is added to the image, scaled by a factor of $\lambda = 0.2$. This process synthesizes an abnormal image $x'$, introducing structured perturbations that mimic potential anomalies.

### 2.3 SCALE-AWARE CONTRASTIVE REVERSE DISTILLATION

To address the scale variation of anomalies in medical images, we propose learning a scale descriptor $\alpha = [\alpha_1, \ldots, \alpha_K]$, where $K$ is the number of layers, using a head $h(\cdot)$. The process involves employing global max pooling (GMP) to spatially shrink $\phi$, generating channel-wise statistics. Since $\phi$ can be viewed as the joint representation of multi-scale features, its statistics are informative about the image content across scales. Subsequently, a linear layer with softmax normalization maps the channel-wise statistics to the scale descriptor $\alpha$. This process is formulated as:

$$\alpha = h(\phi) = \text{softmax}(W \cdot \text{GMP}(\phi)) \, . \tag{3}$$

The final loss is obtained by recalibrating knowledge transfer at different scales with $\alpha$:

$$\mathcal{L} = \sum_k \alpha_k \frac{1 - \text{sim}(u_k, v_k)}{1 - \text{sim}(z_k, v_k) + \epsilon} \, . \tag{4}$$

The scale descriptor $\alpha$ acts as input-specific scale weights for contrastive reverse distillation. In this regard, it intrinsically introduces dynamics conditioned on the input, helping to boost the discriminability of the model.

### 2.4 ANOMALY SCORING

During inference, given a test image, we extract a set of feature maps, $\{u_k\}$ and $\{v_k\}$, from the clean teacher encoder and student decoder, respectively, as defined in Section 2.2.1. Besides, we

obtain a scale descriptor $\alpha$ from the scale adaptation module. We compute the vector-wise cosine similarity between $u_k$ and $v_k$ for each scale $k$. The resulting similarity map is then weighted by the corresponding $\alpha_k$ and upsampled to match the input image resolution. We aggregate weighted similarity maps across all scales to construct a comprehensive anomaly map. The final anomaly score for the input image is obtained by computing the mean value of this aggregated anomaly map.

## 2.5 COMPARISON WITH RD++

A recent work closely related to our approach is RD++ (Tien et al., 2023), which also builds upon the reverse distillation paradigm and utilizes synthesized abnormal images. While the proposed model shares some architectural similarities with RD++, including the use of dual encoding pathways, there are fundamental differences in approach and objectives. A key distinction lies in how anomalous information is used. Our method leverages synthesized anomalies to enhance feature discrimination by employing contrastive regularization between the student's reconstructed features and representations from both teachers. In contrast, RD++ introduces multiple regularizations for two encoding pathways, aiming to encourage the model to learn how to reconstruct normal features from pseudo-abnormal regions. This design inherently restricts the flow of anomalous information to the student network. Furthermore, our approach incorporates a scale adaptation mechanism for reverse distillation, addressing the critical issue of scale variation in anomaly detection—a challenge not explicitly tackled by RD++. We provide a comparison of the two models in Table 1.

# 3 EXPERIMENTS

## 3.1 EXPERIMENTAL SETTINGS

### 3.1.1 DATASETS

We evaluate our proposed method on three widely-used medical imaging datasets: the RSNA Pneumonia Detection Challenge dataset[*], the Brain Tumor MRI dataset[†], and the ISIC 2018 dataset[‡].

**RSNA:** This chest X-ray dataset comprises 8,851 normal and 6,012 lung opacity images. Following Cai et al. (2022), we utilize 3,851 normal images for training and a balanced test set of 1,000 normal and 1,000 abnormal images.

**Brain Tumor:** This dataset consists of 2,000 MRI slices without tumors, 1,621 with gliomas, and 1,645 with meningiomas. We categorize glioma and meningioma slices as anomalies. The normal instances are sourced from Br35H5[§] and Saleh et al. (2020), while the anomalous cases are from Saleh et al. (2020) and Cheng et al. (2015). In line with Cai et al. (2022), our experimental setup includes 1,000 normal slices for training and a test set of 600 normal and 600 abnormal slices (equally split between glioma and meningioma).

**ISIC:** This skin lesion dataset, originating from the ISIC 2018 challenge, contains dermoscopic images across seven categories. Consistent with previous studies (Lu & Xu, 2018; Guo et al., 2024), we designate nevus as the normal class. Our experimental protocol, following Cai et al. (2024), employs 6,705 normal images from the official training set for model training. Our test set comprises 909 normal images and 603 abnormal images (distributed across the remaining six categories) from the official test set.

### 3.1.2 EVALUATION METRICS

Unsupervised anomaly detection methods typically generate continuous-valued predictions. Therefore, we use the area under a receiver operating characteristic (ROC) curve (AUC) as our primary evaluation metric, given its threshold-independent nature. Moreover, we report F1 score and accuracy. For these metrics, we determine the optimal threshold based on the best F1 score, following the approach of Zhao et al. (2023).

---

[*]https://www.kaggle.com/c/rsna-pneumonia-detection-challenge

[†]https://www.kaggle.com/datasets/masoudnickparvar/brain-tumor-mri-dataset

[‡]https://challenge.isic-archive.com/data/#2018

[§]https://www.kaggle.com/datasets/ahmedhamada0/brain-tumor-detection

| | RSNA | | | Brain Tumor | | | ISIC | | |
|---|---|---|---|---|---|---|---|---|---|
| | AUC | F1 | ACC | AUC | F1 | ACC | AUC | F1 | ACC |
| AE | 68.33 | 67.85 | 52.90 | 80.88 | 84.79 | 82.33 | 73.59 | 64.67 | 67.20 |
| UAE | 84.36 | 79.47 | 77.55 | 94.50 | 91.38 | 90.75 | 74.12 | 65.91 | 68.25 |
| MorphAEus | 80.87 | 75.74 | 72.80 | 64.68 | 70.43 | 60.67 | 68.30 | 61.11 | 57.74 |
| GAN Ensemble | 82.10 | 75.30 | 74.30 | 66.60 | 68.40 | 64.00 | 63.70 | 62.70 | 54.60 |
| MemAE | 68.65 | 67.95 | 53.45 | 79.91 | 82.63 | 80.00 | 73.47 | 64.67 | 64.81 |
| PatchCore | 86.33 | 80.86 | 79.40 | 93.63 | 89.23 | 88.42 | 68.09 | 61.98 | 59.26 |
| SQUID | 70.38 | 72.40 | 65.95 | 41.33 | 66.67 | 50.00 | 57.47 | 54.78 | 46.11 |
| FastFlow | 76.00 | 73.68 | 67.95 | 85.62 | 80.41 | 77.58 | 67.27 | 63.46 | 57.34 |
| CFLOW-AD | 70.26 | 70.20 | 62.05 | 36.35 | 66.67 | 50.00 | 66.97 | 60.81 | 57.80 |
| CutPaste | 55.86 | 66.69 | 50.05 | 74.24 | 67.99 | 52.92 | 64.25 | 57.05 | 39.95 |
| NSA | 82.13 | 75.87 | 82.13 | 83.20 | 79.00 | 76.17 | 69.08 | 62.53 | 58.47 |
| RD4AD | 84.29 | 78.09 | 76.80 | 90.52 | 87.24 | 85.67 | 76.48 | 67.94 | 67.72 |
| RD++ | 88.00 | 81.57 | 81.20 | 91.68 | 87.43 | 85.83 | 75.47 | 67.33 | 67.59 |
| ReContrast | 87.53 | 81.24 | 80.70 | 91.67 | 85.99 | 84.33 | 80.02 | 70.20 | 75.46 |
| UniAD | 73.69 | 69.71 | 65.85 | 62.30 | 69.92 | 58.42 | 73.78 | 65.48 | 66.60 |
| SimpleNet | 69.06 | 68.99 | 62.70 | 93.93 | 88.74 | 88.50 | 69.01 | 60.66 | 56.99 |
| EfficientAD | 74.88 | 73.12 | 68.20 | 78.41 | 76.20 | 72.00 | 60.32 | 56.95 | 39.81 |
| UCAD | 70.89 | 69.88 | 62.45 | 87.42 | 80.68 | 80.08 | 67.88 | 59.62 | 55.29 |
| Ours | **91.01** | **84.05** | **83.40** | **98.88** | **96.52** | **96.50** | **83.10** | **72.07** | **75.60** |

Table 1: Quantitative comparison of the proposed method against state-of-the-art approaches on three public medical anomaly detection datasets. The best results for each dataset and metric are highlighted in **bold**, and the second-best results are underlined.

### 3.1.3 IMPLEMENTATION DETAILS

We conduct all experiments using PyTorch on a single NVIDIA RTX 3090Ti GPU. Our encoder utilizes a WideResNet50 (Zagoruyko, 2016) pre-trained on ImageNet (Russakovsky et al., 2015). We also report the performance of our network using ResNet18 (He et al., 2016) and ResNet50 (He et al., 2016) as the encoder in Section 3.3.4. We resize all images to $256 \times 256$ pixels and apply no data augmentation during training. To train our model, we employ the Adam optimizer (Kingma & Ba, 2015) with $\beta = (0.5, 0.999)$ and a learning rate of 1e-3. We train for 4,000 iterations with a batch size of 16. Our decoder mirrors the encoder, identical to that used in RD4AD (Deng & Li, 2022). For competing methods, we utilize their publicly available codes and adhere to their default training configurations.

### 3.2 COMPARISON WITH STATE-OF-THE-ART METHODS

We comprehensively evaluate our approach against existing unsupervised anomaly detection methods. Our comparisons encompass reverse distillation-based methods such as RD4AD (Deng & Li, 2022) and its variants, RD++ (Tien et al., 2023) and ReContrast (Guo et al., 2023), as the proposed model builds upon this paradigm. Given our use of synthetic anomalies, we also compare with synthetic anomaly-based methods like NSA (Schlüter et al., 2022) and CutPaste (Li et al., 2021). Furthermore, we assess our approach against methods using generative models (AE, UAE (Mao et al., 2020), GAN Ensemble (Han et al., 2021b), and MorphAEus (Bercea et al., 2023)), methods utilizing memory banks (MemAE (Gong et al., 2019), PatchCore (Roth et al., 2022), and SQUID (Xiang et al., 2023)), and methods leveraging normalizing flows (FastFlow (Yu et al., 2021) and CFLOW-AD (Gudovskiy et al., 2022)). We also include other recent relevant methods such as UniAD (You et al., 2022), SimpleNet (Liu et al., 2023), EfficientAD (Batzner et al., 2024), and UCAD (Liu et al., 2024) in our comparison. As shown in Table 1, the proposed framework demonstrates superior performance across all three datasets, consistently outperforming competitors on all evaluation metrics. Notably, in terms of the primary metric AUC, our method shows substantial improvements over the second-best methods. On the RSNA dataset, we achieve a 3.01% improvement. For the Brain

| CRD | SAM | RSNA | | | Brain Tumor | | | ISIC | | |
|-----|-----|------|------|------|------|------|------|------|------|------|
| | | AUC | F1 | ACC | AUC | F1 | ACC | AUC | F1 | ACC |
| - | - | 84.29 | 78.09 | 76.80 | 90.52 | 87.24 | 85.67 | 76.48 | 67.94 | 67.72 |
| ✓ | - | 89.87 | 83.69 | 83.20 | 91.45 | 88.17 | 86.67 | 78.87 | 69.50 | 71.56 |
| ✓ | ✓ | **91.01** | **84.05** | **83.40** | **98.88** | **96.52** | **96.50** | **83.10** | **72.07** | **75.60** |

Table 2: Ablation study quantifying the impact of each component in the proposed method on three datasets. We report the performance of our full model, as well as variants with the following components ablated: contrastive reverse distillation (CRD) and scale-adaptive mechanism (SAM).

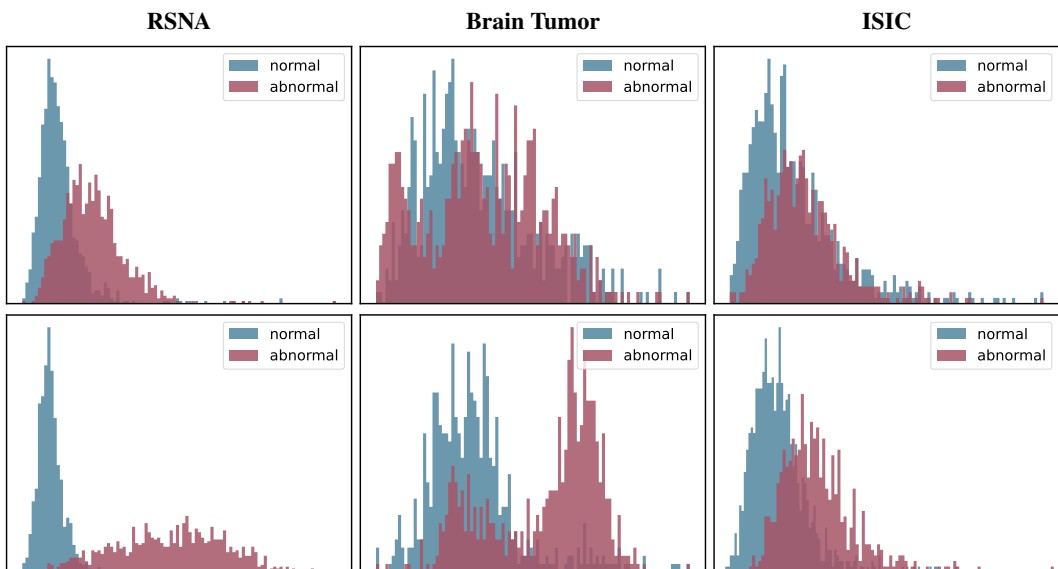

Figure 2: Comparison of anomaly score distributions for normal (blue) and abnormal (red) samples in the test sets across datasets. Top: Distributions obtained from the baseline RD4AD (Deng & Li, 2022). Bottom: Distributions generated by our proposed model. Scores are normalized to [0,1] for each subfigure to enable direct comparison. Our approach demonstrates enhanced separation, leading to improved anomaly detection performance.

Tumor dataset, the improvement is 4.38%, while on the ISIC dataset, we see a 3.08% increase in performance.

## 3.3 ABLATION STUDY

We conduct ablation studies to validate the efficacy of each component in our framework. Using the RD4AD model (Deng & Li, 2022) as our baseline, we progressively incorporate our contributions, including contrastive reverse distillation and the scale-adaptive mechanism. Table 2 presents the numerical results.

### 3.3.1 EFFECTIVENESS OF CONTRASTIVE REVERSE DISTILLATION

Augmenting the reverse distillation model with the proposed contrastive student-teacher learning yields significant improvements. On the RSNA dataset, we achieve gains of 5.58%, 5.6%, and 6.4% in AUC, F1 score, and accuracy, respectively, over the baseline. We observe consistent improvements across all metrics on the Brain Tumor and ISIC datasets as well.

### 3.3.2 EFFECTIVENESS OF SCALE ADAPTIVE MECHANISM

Further incorporating the scale adaptive mechanism into the contrastive reverse distillation framework leads to additional performance enhancements. On the Brain Tumor dataset, we observe improvements of 7.43%, 8.35%, and 9.83% in AUC, F1 score, and accuracy, respectively. The RSNA dataset shows consistent gains (1.14% in AUC, 0.36% in F1 score, and 0.2% in accuracy), as does the ISIC dataset (4.23% in AUC, 2.57% in F1 score, and 4.04% in accuracy). These results confirm the importance of addressing scale variation in anomaly detection for medical images.

### 3.3.3 IMPACT OF NOISE TYPE AND INTENSITY

We evaluate our model's performance using two types of noise for creating synthetic anomalies: simplex noise and Gaussian noise. Table 3 presents the results, indicating that simplex noise generates more natural pseudo-anomalies that better align with anatomical structures in medical imaging compared to Gaussian noise.

We further investigate the impact of varying simplex noise intensities on model performance. Figure 3 shows our method's performance on the RSNA dataset as $\lambda$ increases from 0.1 to 0.5. We observe that the intensity of simulated anomalies significantly influences model performance. Low noise ($\lambda = 0.1$) appears insufficient to effectively emulate real abnormalities in medical images. The model's performance peaks at a moderate intensity ($\lambda = 0.2$). When $\lambda$ reaches 0.5, our approach's performance drops sharply, suggesting that excessive noise may interfere with key medical features, making it challenging for the model to distinguish between real pathological changes and excessive image distortions.

The results demonstrate that the choice of noise type and intensity plays a crucial role in the effectiveness of synthetic anomaly generation for improving anomaly detection in the proposed framework.

### 3.3.4 QUANTITATIVE COMPARISON ACROSS DIFFERENT BACKBONES

Table 4 presents the performance of our model with various backbone architectures. Generally, deeper and wider networks exhibit stronger representational capabilities, enhancing our model's ability to detect anomalies more precisely.

### 3.3.5 QUALITATIVE ANALYSIS

To qualitatively assess the discriminative capability of our framework, we visualize the distributions of anomaly scores for normal and abnormal images within all datasets in Figure 2. The overlap between normal and abnormal histograms represents samples from different categories that share identical anomaly scores. A smaller overlap indicates a stronger discriminative capability of a model in separating normal and abnormal instances. As evident from Figure 2, compared to the baseline model, the proposed framework better separates normal and abnormal images, exhibiting superior anomaly detection performance.

## 3.4 DISCUSSION

The performance difference between ISIC and Brain Tumor datasets arises from their inherent structural complexities. The ISIC dataset contains multiple dermatological anomalies (melanoma, melanocytic nevus, basal cell carcinoma, actinic keratosis, benign keratosis, dermatofibroma, and vascular lesion) with subtle, often indistinct morphological features, challenging model discrimination. In contrast, Brain Tumor images reveal well-defined, high-contrast lesions with clear structural boundaries, rendering anomaly detection more straightforward for models.

The Brain Tumor MRI dataset features highly variable tumor sizes across slices, making the SAM module more impactful, whereas the RSNA X-ray dataset exhibits more consistent anomaly sizes, thus limiting SAM's comparative effectiveness. This differential performance aligns with the underlying dataset characteristics and underscores the importance of scale adaptability across diverse medical imaging contexts.

| Noise Type | RSNA | | | Brain Tumor | | | ISIC | | |
|---|---|---|---|---|---|---|---|---|---|
| | AUC | F1 | ACC | AUC | F1 | ACC | AUC | F1 | ACC |
| Gaussian | 90.70 | **84.11** | **84.05** | 75.23 | 74.88 | 68.25 | 74.30 | 66.20 | 67.99 |
| Simplex | **91.01** | 84.05 | 83.40 | **98.88** | **96.52** | **96.50** | **83.10** | **72.07** | **75.60** |

Table 3: Comparative analysis of model performance under various noise types on all datasets.

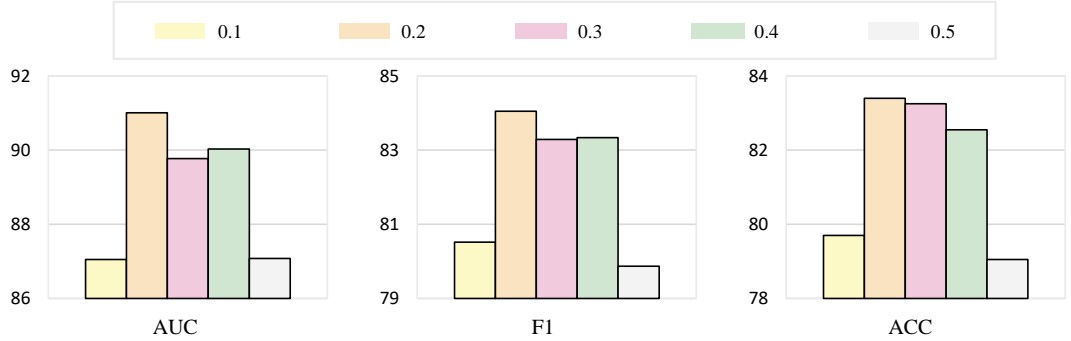

Figure 3: Effect of $\lambda$ on model performance on the RSNA dataset. $\lambda$ ranges from 0.1 to 0.5 in 0.1 increments, with higher values corresponding to increased simplex noise levels. Performance metrics (AUC, F1, and ACC) are shown as $\lambda$ increases left to right.

## 4    RELATED WORK

Reconstruction-based methods have emerged as a prominent approach in unsupervised anomaly detection. Schlegl et al. (2017) pioneer the use of GANs for this purpose with AnoGAN, later introducing f-AnoGAN (Schlegl et al., 2019), a faster variant employing an encoder to map images to a latent space. In addition, various autoencoder architectures are explored, including variational autoencoder (Zimmerer et al., 2018) and vector-quantized variational autoencoder (Naval Marimont & Tarroni, 2021). To address the overgeneralization problem, Gong et al. (2019) propose a memory-augmented autoencoder. Given an input image, they first obtain an encoded representation from the encoder and then use it as a query to retrieve the most relevant memory items for reconstruction. Park et al. (2020) exploit a memory module to record prototypical patterns of normal instances. Several works (Rudolph et al., 2021; Gudovskiy et al., 2022; Yu et al., 2021) leverage normalizing flows, enabling exact likelihood estimation for image modeling, and achieve good performance in anomaly detection. For example, Gudovskiy et al. (2022) use a conditional normalizing flow framework, and Yu et al. (2021) propose a 2D normalizing flow model for unsupervised anomaly detection and localization.

Self-supervised learning (Jing & Tian, 2021) has also been applied to anomaly detection. Some works train models to detect synthetic anomalies and directly apply them to real abnormalities. To make synthesized anomaly images more natural, Tan et al. (2021) and Schlüter et al. (2022) integrate Poisson image editing to seamlessly blend scaled patches of various sizes from separate images. This creates a wide range of synthetic anomalies that are more similar to natural abnormalities than previous data augmentation strategies for self-supervised anomaly detection. Furthermore, two-stage approaches are studied. For instance, Li et al. (2021) propose a simple strategy to generate synthetic anomalies for anomaly detection by cutting an image patch and pasting it at a random location of a large image. They then learn self-supervised representations by classifying normal and sythesized abnormal data. Finally, a generative one-class classifier is built on the learned representations. Sohn et al. (2021) learn self-supervised representations from normal data by solving proxy tasks, e.g., rotation prediction and contrastive learning, and then train one-class classifiers using the learned representations.

Knowledge distillation from pre-trained models showcases promising results recently (Salehi et al., 2021; Deng & Li, 2022; Batzner et al., 2024). Salehi et al. (2021) propose to use the distillation

| Backbone | RSNA | | | Brain Tumor | | | ISIC | | |
|---|---|---|---|---|---|---|---|---|---|
| | AUC | F1 | ACC | AUC | F1 | ACC | AUC | F1 | ACC |
| Resnet-34 | 85.20 | 78.96 | 77.25 | 92.17 | 87.95 | 86.67 | 78.76 | 70.43 | 69.44 |
| Resnet-50 | 88.45 | 82.15 | 81.25 | 94.16 | 89.81 | 89.00 | 80.83 | 71.28 | 71.96 |
| Wide ResNet-50 | **91.01** | **84.05** | **83.40** | **98.88** | **96.52** | **96.50** | **83.10** | **72.07** | **75.60** |

Table 4: Quantitative performance comparison of multiple backbone architectures for all datasets.

of features at various layers of an expert network, pre-trained on ImageNet, into a simpler cloner network. They detect anomalies using the discrepancy between the expert and cloner networks' intermediate feature maps given an input image. Deng & Li (2022) devise a reverse distillation paradigm, which is further explored in subsequent works (Deng & Li, 2022; Tien et al., 2023).

Recently, You et al. (2022) formulate universal anomaly detection and propose a Transformer-based feature reconstruction model using a layer-wise query decoder to model complex multi-class normal distributions.

## 5 CONCLUSION

This paper presents a novel scale-aware contrastive reverse distillation model for unsupervised anomaly detection. Our approach introduces a contrastive student-teacher framework comprising a clean teacher encoder, a noisy teacher encoder, and a student decoder, coupled with a scale adaptation mechanism. This architecture enables our model to derive robust feature representations and effectively address the scale variation issue inherent in anomalies. Extensive experiments on benchmark datasets demonstrate that the proposed method achieves state-of-the-art performance, underscoring its efficacy in unsupervised anomaly detection tasks. Our current approach relies on random spatial sampling for noise generation. Future research directions include incorporating modality-specific anatomical priors for anomaly localization in medical imaging. The exploration of learnable noise like (Cai & Fan, 2022) or adaptive hybrid noise generation techniques presents another promising direction. These extensions would enhance the realism of synthetic anomalies toward improved model performance.

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

APPENDIX

**Algorithm.** We present detailed procedures for synthesizing abnormal images and training our model in Algorithm 1 and Algorithm 2, respectively.

---

**Algorithm 1** Synthesize abnormal images
---

$h$ is the height of the input image.
$w$ is the width of the input image.
$[a, b]$ represents the set $\{x \in \mathbb{R} : a \leq x \leq b\}$
**for** epoch = 1 to $n$ **do**
    **for** $\boldsymbol{x}_i$ in normal training set **do**
        Get $h_{noise} \subseteq [10, \text{int}(h/8)]$
        Get $w_{noise} \subseteq [10, \text{int}(w/8)]$
        Get $x_{start}, y_{start} \subseteq [0, h - h_{noise}], [0, w - w_{noise}]$
        Randomly generate simplex noise:
        $\boldsymbol{\epsilon} \sim \text{Simplex}((h_{noise}, w_{noise}), N = 6, \gamma = 0.6)$
        $\boldsymbol{\xi} = \text{zeros}(h, w)$
        $x_{end} = x_{start} + h_{noise}$
        $y_{end} = y_{start} + w_{noise}$
        $\boldsymbol{\xi}[x_{start} : x_{end}, y_{start} : y_{end}] = \boldsymbol{\epsilon}$
        $\boldsymbol{x}_i' = \boldsymbol{x}_i + \lambda * \boldsymbol{\xi}$   ($\lambda$: the intensity of the added noise)
        Training process
    **end for**
**end for**

---

---

**Algorithm 2** Pseudo-code of our approach in one epoch training
---

$\mathcal{E}, \mathcal{S}, \mathcal{B}, \mathcal{D}$: Encoder, Scale Adaptation Module, Bottleneck, Decoder
$\boldsymbol{u}_k, \boldsymbol{z}_k$: Normal and synthetic anomalous features at block $k$
Optimizer = Adam
Load a mini-batch of normal and pseudo-abnormal samples
**for** $\boldsymbol{x}, \boldsymbol{x}'$ in train-dataloader **do**
    Get encoder outputs for normal and pseudo-abnormal images at three blocks
    $\boldsymbol{u}_1, \boldsymbol{u}_2, \boldsymbol{u}_3 = \mathcal{E}(\boldsymbol{x})$
    $\boldsymbol{z}_1, \boldsymbol{z}_2, \boldsymbol{z}_3 = \mathcal{E}(\boldsymbol{x}')$
    Get decoder outputs
    $\boldsymbol{v}_1, \boldsymbol{v}_2, \boldsymbol{v}_3 = \mathcal{D}(\mathcal{B}(\boldsymbol{u}_1, \boldsymbol{u}_2, \boldsymbol{u}_3))$
    $\boldsymbol{\alpha} = \mathcal{S}(\mathcal{B}(\boldsymbol{u}_1, \boldsymbol{u}_2, \boldsymbol{u}_3))$
    $\mathcal{L}_1 = \sum_{k=1}^{3} \alpha_k (1 - \text{sim}(\boldsymbol{u}_k, \boldsymbol{v}_k))$
    $\mathcal{L}_2 = \sum_{k=1}^{3} \alpha_k (1 - \text{sim}(\boldsymbol{z}_k, \boldsymbol{v}_k))$
    $\mathcal{L} = \mathcal{L}_1 / \mathcal{L}_2$
    $\mathcal{L}$.backward
    Optimizer.step
**end for**

---

**Inference Time and Paramters.** We compare our model with competing methods in terms of AUC, inference time, and memory usage at inference (see Figure 4). The focus on the inference phase is particularly relevant due to its critical importance in clinical applications, where real-time processing and memory efficiency directly impact the feasibility and deployment potential of a model. The advantages of our model in these aspects make it a promising approach for practical applications. We additionally report trainable parameter counts for our model and competing methods in Table 5.

**ROC Curves.** We visualize the ROC curves to compare our method with the top-performing baselines across all datasets (see Figures 5, 6, 7).

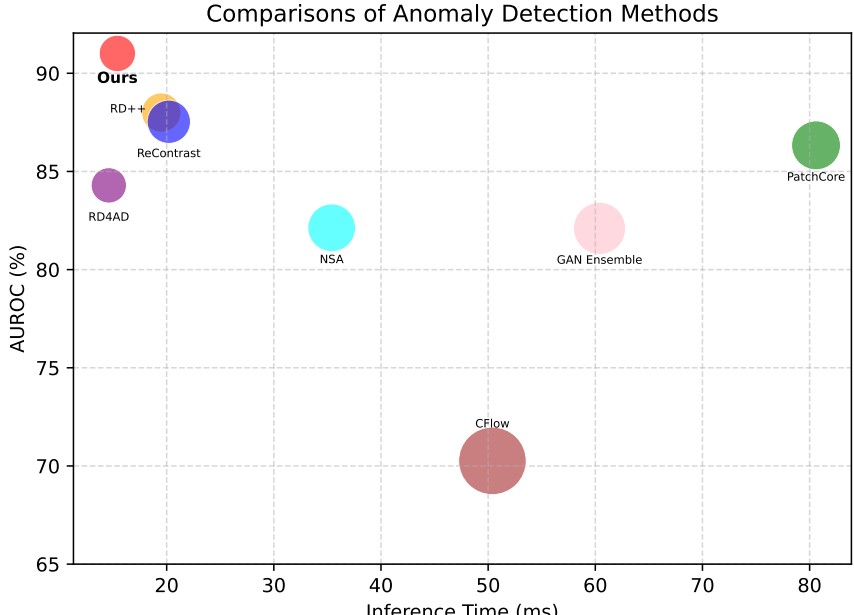

Figure 4: Performance comparison of anomaly detection methods. Evaluation metrics include: AUROC (vertical axis), inference time (horizontal axis), and memory footprint (circle radius). Our method achieves state-of-the-art performance, delivering the highest AUROC while demonstrating superior computational efficiency. Specifically, our approach is 6x faster than PatchCore, 4x faster than GAN Ensemble, 3x faster than CFlow, and 2x faster than NSA.

|  | Ours | RD4AD | RD++ | ReContrast | CFlow | GAN Ensemble | PatchCore | UAE | NSA |
|---|---|---|---|---|---|---|---|---|---|
| **AUC** | 91.01 | 84.29 | 88.00 | 87.53 | 70.26 | 82.10 | 86.33 | 84.36 | 82.13 |
| **Params (MB)** | 264.4 | 263.5 | 272.8 | 527.9 | 288.5 | 381.46 | 275.6 | 256.3 | 271.2 |

Table 5: Comparative performance of anomaly detection methods on the RSNA dataset: AUC scores and model complexity. Our approach achieves the highest AUC with a more parameter-efficient design.

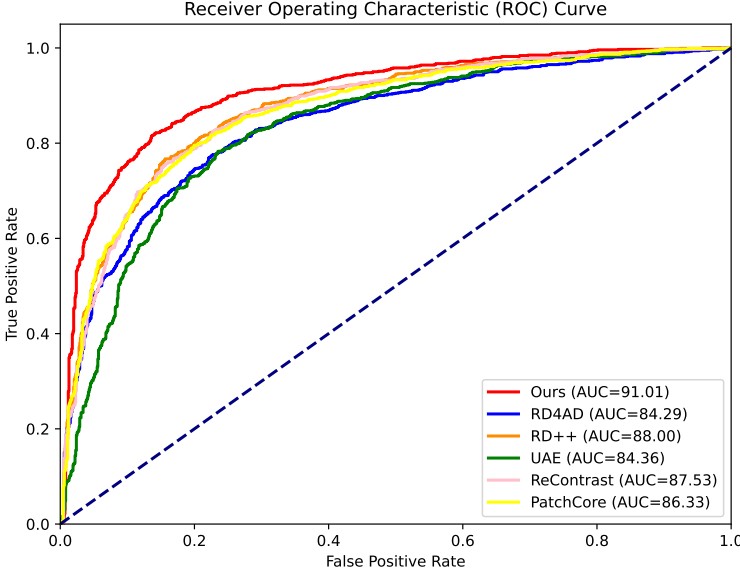

Figure 5: Comparison of ROC curves between our method and the top 5 anomaly detection methods on the RSNA dataset.

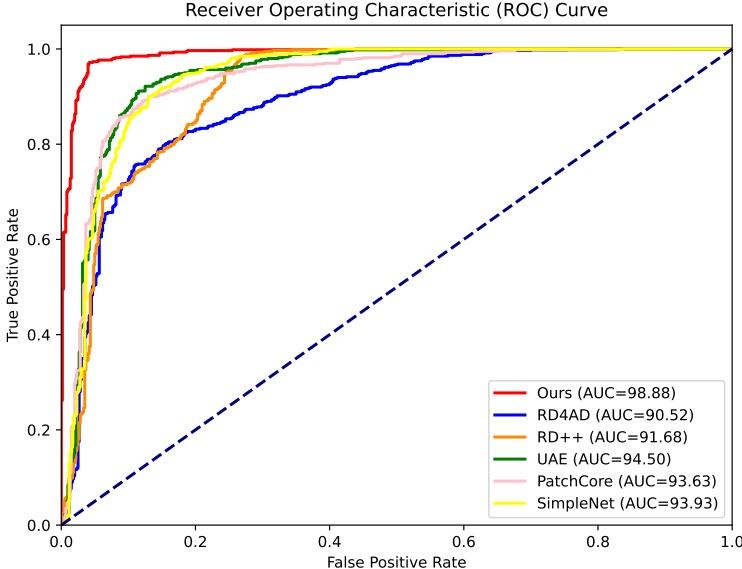

Figure 6: Comparison of ROC curves between our method and the top 5 anomaly detection methods on the Brain Tumor dataset.

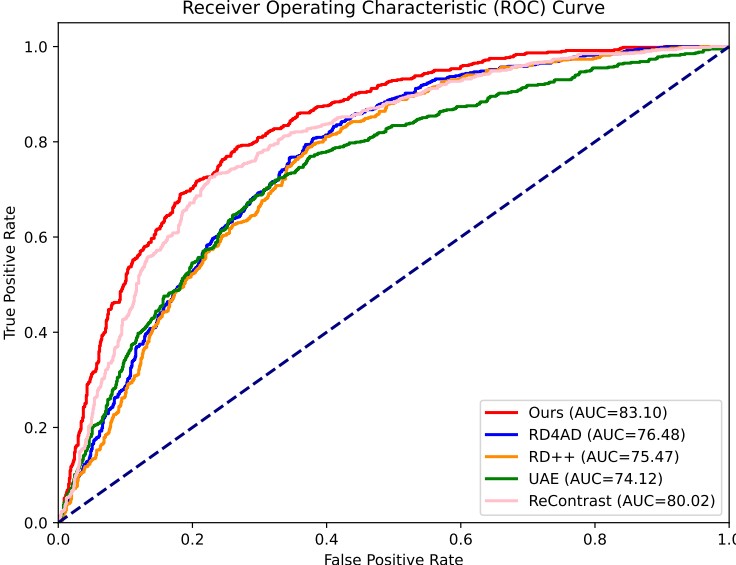

Figure 7: Comparison of ROC curves between our method and the top 4 anomaly detection methods on the ISIC dataset.

