# OpenReview forum: "Scale-Aware Contrastive Reverse Distillation for Unsupervised Medical Anomaly Detection"
_ICLR.cc/2025/Conference — ICLR 2025 Poster_

### Official Review · Reviewer_8DJE · 2024-10-18

**Soundness:** 3
**Presentation:** 3
**Contribution:** 3
**Rating:** 8
**Confidence:** 3

**Summary:**

The authors propose a new method for anomaly detection in medical data setting using contrastive reverse distillation. The method uses knowledge distillation as it's foundation, implementing pre-trained teachers models along with a student model trained to either minimize or maximize its distance from the teachers in a feature-map space. They additionally utilize a scale adaptive mechanism to allow for better generalizations across different sizes of anomaly.

Their method's specific construction utilizes two teacher models with the same weights, one "bad" and one "good". The good model is given benign images while the bad model is given images with synthetic anomalies. The student model is then trained to maximize its cosine similarity with the per-layer features of the good teacher while minimizing the cosine similarity with the bad teacher. At evaluation time anomaly scores are computed by comparing the scaled, per-layer feature spaces of the good teacher against those of the student. Inputs with high levels of dissimilarity between the good model and student are given high anomaly scores.

Their evaluation against prior work is very comprehensive, comparing against a total of 18 other detectors. They out perform all other detectors on all three datasets (lung, brain, and skin anomalies) in terms of their main metrics (AUC, F1, and Accuracy). The authors further perform ablations with respect to their main contributions of Contrastive Reverse Distilation and Scale Adaptive Mechanisms.

**Strengths:**

* Anomaly detection is somewhat outside my area of expertise, but the paper seems to frame itself well within the current body of literature.
* The paper is well written and easy to follow. Figures are also informative and easy to understand.
* The idea of contrastive reverse distillation seems novel (to the best of my knowledge) and intuitive. I'd like to see how future works can iterate upon this framework in other settings or with improved constructions.
* Motivation is well established for the usage synthetic anomalies, and the choice of simplex noise is clever.
* Empirical analysis is very comprehensive in terms of datasets, baselines, and ablations.
* The proposed method out-performs all baselines in terms of AUC, F1, and Accuracy.
* The proposed method scales well with model architecture size, which is a good sign that the method is sensible.

**Weaknesses:**

In general, within my somewhat limited knowledge of this sub-field, I like this paper. In spite of that I think there are a few things the authors may want to consider to further strengthen their work.

I think when discussing detection and AUC scores it is important to give them context within the given domain. See section B of [1] for some example discussion on this topic from a different field. I'd like the authors to consider the following questions:

* Is it more important to maximize the true positive rate in spite of inducing some false positives?
* Is it better to minimize false positives to increase confidence in the case of a positive prediction?
* How do you anticipate doctors using these tools in their analysis?

Given this, it would also be nice if the authors could plot some of the ROCs between their method and some of the baselines (maybe including all of them is too visually messy, but at least the top 5). This way people can compare their method to the baselines at different levels of false positive rate.

I think it would be good if the authors could include some extended discussion, either in the main body or the supplementary material, analyzing the more subtle features of their results. See the next section of my review "Questions" for some examples of what I think they can further analyze.

Lastly, the results of table 1 are good, but a bit overwhelming. I think it would help if the second best score in each metric is underlined and if the methods are visually grouped based on detector type (e.g. RD4AD, RD++, and ReContrast could be grouped together). Optionally it would be nice if all experiments were averaged over a few runs and given standard deviations or $\pm$ scores, at least in the supplementary material. I understand if these experiments are too extensive to replicate too many times, though.

 [1] "Membership Inference Attacks From First Principles" https://arxiv.org/abs/2112.03570

**Questions:**

Here I include some questions for the authors to consider.

* Why does ISIC seem like the hardest dataset for all the detectors? Why is Brain Tumor the easiest?
* Why is SAM more impactful in the Brain Tumor dataset while it seems minimally impactful in the RSNA dataset in table 2? Is this related at all to the results of table 3 - where Gaussian noise performs similar to simplex noise on RSNA, but much worse on Brain Tumor?
* Can the method achieve better generalization by using a mixture of synthetic noise types, perhaps 10% gaussian, 90% simplex?
* Can we improve performance by including some real world data in the training set, or will that harm generalization?

---

> ### Author Response · Authors · 2024-11-25
>
> > Given this, it would also be nice if the authors could plot some of the ROCs between their method and some of the baselines (maybe including all of them is too visually messy, but at least the top 5). This way people can compare their method to the baselines at different levels of false positive rate.
>
> Many thanks for the suggestion regarding the visualization of ROC curves. We have now added ROC curve plots comparing our method against the top-performing baselines across all datasets (see Figures 5,6,7 **in the appendix**).

---

> ### Author Response · Authors · 2024-11-25
>
> > I think it would be good if the authors could include some extended discussion, either in the main body or the supplementary material, analyzing the more subtle features of their results. See the next section of my review "Questions" for some examples of what I think they can further analyze.
>
> >>Why does ISIC seem like the hardest dataset for all the detectors? Why is Brain Tumor the easiest?
>
> The performance difference between ISIC and Brain Tumor datasets arises from their inherent structural complexities. The ISIC dataset contains multiple dermatological anomalies (melanoma, melanocytic nevus, basal cell carcinoma, actinic keratosis, benign keratosis, dermatofibroma, and vascular lesion) with subtle, often indistinct morphological features, challenging model discrimination. In contrast, Brain Tumor images reveal well-defined, high-contrast lesions with clear structural boundaries, rendering anomaly detection more straightforward for models.
>
> ---
>
> >> Why is SAM more impactful in the Brain Tumor dataset while it seems minimally impactful in the RSNA dataset in table 2? Is this related at all to the results of table 3 - where Gaussian noise performs similar to simplex noise on RSNA, but much worse on Brain Tumor?
>
> The Brain Tumor MRI dataset features highly variable tumor sizes across slices, making the SAM module more impactful, whereas the RSNA X-ray dataset exhibits more consistent anomaly sizes, thus limiting SAM's comparative effectiveness. This differential performance aligns with the underlying dataset characteristics and underscores the importance of scale adaptability across diverse medical imaging contexts.
>
> ---
>
> >> Can the method achieve better generalization by using a mixture of synthetic noise types, perhaps 10% gaussian, 90% simplex?
>
> We appreciate the reviewer's insightful suggestion regarding noise type mixture. While we have not empirically validated this specific approach in our current submission, the proposed strategy of combining different synthetic noise types represents a promising direction for future research. Our paper indeed identifies noise robustness as a key area for potential method enhancement, and the reviewer's specific suggestion offers a concrete pathway for exploring improved generalization. In our ongoing work, we plan to systematically investigate such noise mixture strategies to understand their potential impact on model performance and robustness. We discuss this in the revised version of the submission.
>
> ---
>
> >> Can we improve performance by including some real world data in the training set, or will that harm generalization?
>
> We appreciate the insightful suggestion regarding real-world data incorporation. While introducing real-world anomalous data could potentially improve performance, doing so would fundamentally alter the problem setting from unsupervised to supervised/semi-supervised anomaly detection. Our current work specifically focuses on unsupervised anomaly detection, and therefore we deliberately maintained the original task constraints.

---

> ### Author Response · Authors · 2024-11-25
>
> >  I think it would help if the second best score in each metric is underlined and if the methods are visually grouped based on detector type (e.g. RD4AD, RD++, and ReContrast could be grouped together). Optionally it would be nice if all experiments were averaged over a few runs and given standard deviations or scores, at least in the supplementary material. [...]
>
> We thank the reviewer for these constructive suggestions.
>
> 1. As suggested, we have revised Table 1 to improve its readability by visually grouping the methods based on methodology and underlining the second-best score for each metric on each dataset.
>
>
> 2. Regarding the suggestion to report means and standard deviations across multiple runs, we appreciate the scientific rigor of this request. However, given the computational intensity of these experiments across 19 methods and the limited time available during the review period, we regrettably cannot complete multiple runs at this time. We acknowledge the importance of this analysis and commit to conducting these additional experiments, with the results to be included in the appendix of the final version.

---

> > ### Comment · Reviewer_8DJE · 2024-11-25
> > **Thank You**
> >
> > Thanks for the detailed response and for updating the paper with my suggestions. I think it may help improve the paper for readers outside of the medical field if you add some of the context you explained here (e.g. about ISIC vs brain tumor dataset) to the main body of the paper, but I'll leave that decision up to you all.

---

> > > ### Author Response · Authors · 2024-11-27
> > >
> > > As suggested, we have added the recommended context to the main body of the paper.
> > >
> > > We gratefully acknowledge the reviewer's insightful feedback, which has significantly strengthened our manuscript.

---

### Official Review · Reviewer_tPrf · 2024-10-26

**Soundness:** 3
**Presentation:** 4
**Contribution:** 2
**Rating:** 8
**Confidence:** 3

**Summary:**

Two issues arise in reverse knowledge distillation for medical imaging, an inability to properly distinguish between different features and an inability to deal with different scales. This work proposes two methods to address these issues: a contrastive student-teacher learning approach that involves using both a “good” and a “bad” teacher, as well as a scale adaptation mechanism. Results demonstrate that on almost every dataset the proposed approach is able to outperform existing techniques.

**Strengths:**

The paper is well written and the ideas are easy to follow. I appreciate how the authors identify the two problems facing medical imaging in this domain and then propose solutions. The new approach outperforms all other methods that the authors compare with in table 1.

**Weaknesses:**

Issue 1: This paper doesn’t have any formal proofs or theorems to guarantee its effectiveness, therefore the strengths of the paper must rely on the experiments and empirical results. For the datasets I will admit as a review I am not very familiar with them. For example, on the image dataset CIFAR-10, a few years ago a jump from 93% to 98% was considered significant. Today an increase from 99% to 99.9% on CIFAR-10 would not even be considered a major contribution. Likewise, for these datasets it is hard to determine whether a jump of 3.01% and 4.38% (as the authors show) is worthy of publication. Could the authors please clarify how much other papers pushed the accuracy forward in other works, as compared to the increase their method is offering?

Issue 2: The authors say they will release their code upon publication, but I did not see any code given in the supplemental material. This might be an oversight on my part, but where is the code? I would like to see a good faith gesture of releasing an anonymous github for the code or at least a zip file of the code. Too many papers claim SOTA results and never release code. In this day and age it is simply unacceptable to continue this trend.

Issue 3: The authors also don’t mention anything about the runtime of their algorithm. How does the student-teacher setup and training time compare to other existing methods in the field? While this is not a HUGE concern, I think it would be good to have some discussion and analysis of this. For example could you show the training time complexity of your algorithm and the next two best methods in the field? Or even an experimental runtime of the training in seconds of your algorithm and the next two best methods? Without this I feel we don’t have the complete story when comparing methods.

Issue 4: Please see my minor comments about uncited claims in the papers. For acceptance I would like to see citations for all the claims you have made.

Issue 5: I don’t like the terminology “good” and “bad” teacher. It is very non-technical and hard to follow. Can you please change the terminology in your paper and name the teachers more appropriately?

Minor Comments:

=Line 36, “anomalous samples is often prohibitively expensive and time-consuming…” I believe this is correct but I want to see a citation to backup this claim?

=Line 45, “they tend to generalize too well, thereby risking the reconstruction of abnormal regions.” Citation for this claim?

=Table 1 has many entries that are unsorted (aside from the bolded number indicating the best method). It would be better if this table was sorted in order of increasing AUC for the methods.

=Line 167 “Medical images typically exhibit a power law distribution of frequencies, with lower frequency components dominating the image content.” Please give a citation for this bold claim?

**Questions:**

Q1: Please address each of my issues 1-5. If they are adequately addressed I would be willing to increase my score for the paper accordingly.

EDIT: The reviewers have address my concerns adequately. I have update the score of my review accordingly.

---

> ### Author Response · Authors · 2024-11-25
>
> >This paper doesn’t have any formal proofs or theorems to guarantee its effectiveness, therefore the strengths of the paper must rely on the experiments and empirical results. For the datasets I will admit as a review I am not very familiar with them. [...]
>
> We thank the reviewer for this important question regarding the significance of our improvements. To contextualize our gains, we have provided a comparison with recent works evaluated on the same medical imaging datasets. For instance, [1] reports improvements of 1.6% and 0.2% on the RSNA and Brain Tumor datasets respectively, while [2] demonstrates a gain of 0.6% on the RSNA dataset. [3] and [4] show improvements of 2.36% and 2.74% on the ISIC dataset, respectively. In this context, our method’s improvements of 3.01% , 4.38%, and 3.08% on the RSNA, Brain Tumor, and ISIC datasets represent significant progress in the field. Unlike CIFAR-10, where performance is already near saturation, medical imaging datasets still present significant challenges where such margins of improvement can meaningfully impact clinical applications. Our gains, being notably larger than those reported in contemporary works, demonstrate the effectiveness of our approach.
>
> [1] Yu Cai, Hao Chen, Xin Yang, Yu Zhou, and Kwang-Ting Cheng. Dual-distribution discrepancy with self-supervised refinement for anomaly detection in medical images. Medical Image Analysis, 86:102794, 2023.
>
> [2] Yu Cai, Hao Chen, Xin Yang, Yu Zhou, and Kwang-Ting Cheng. Dual-distribution discrepancy for anomaly detection in chest X-Rays. In MICCAI, 2022.
>
> [3] Jia Guo, Shuai Lu, Lize Jia, Weihang Zhang, and Huiqi Li. ReContrast: Domain-specific anomaly detection via contrastive reconstruction. In NeurIPS, 2023.
>
> [4] Jia Guo, Shuai Lu, Lize Jia, Weihang Zhang, and Huiqi Li. Encoder-decoder contrast for unsupervised anomaly detection in medical images. Institute of Electrical and Electronics Engineers Transactions on Medical Imaging, 43(3):1102–1112, 2024.
>
> >The authors say they will release their code upon publication, but I did not see any code given in the supplemental material. This might be an oversight on my part, but where is the code? [...]
>
> We sincerely appreciate the reviewer’s concern regarding code availability. We fully agree that sharing code is crucial for research transparency and reproducibility in our field. In response to this feedback, we have created an anonymous GitHub repository containing our complete implementation, which can be accessed at https://anonymous.4open.science/r/Scale-Aware-Contrastive-Reverse-Distillation-for-Unsupervised-Anomaly-Detection-65E1/. We thank the reviewer for his/her patience and for bringing this important point to our attention.

---

> > ### Author Response · Authors · 2024-11-25
> >
> > >The authors also don’t mention anything about the runtime of their algorithm. How does the student-teacher setup and training time compare to other existing methods in the field? [...]
> >
> > First, following [1], we compare our model with competing methods in terms of AUC, inference time, and memory usage at inference (see Figure 4 **in the appendix**). The focus on the inference phase is particularly relevant due to its critical importance in clinical applications, where real-time processing and memory efficiency directly impact the feasibility and deployment potential of a model.
> >
> > [1] Tran Dinh Tien, Anh Tuan Nguyen, Nguyen Hoang Tran, Ta Duc Huy, Soan Thi Minh Duong,
> > Chanh D. Tr. Nguyen, and Steven Q. H. Truong. Revisiting reverse distillation for anomaly detection. In CVPR, 2023.
> >
> > Regarding training complexity, while our model requires additional computational time during training due to the CPU-based noise generation step, we have opted to present a more standardized comparison through the number of trainable parameters. The updated parameter counts for our model and competing approaches are now included in the table below and **in the appendix** (see Table 5). We believe that parameter count comparison provides a more consistent measure of training complexity across different hardware configurations. Notably, our model achieves state-of-the-art performance across multiple medical imaging datasets while maintaining a relatively compact parameter count, demonstrating an effective balance between model efficiency and performance.
> >
> > | Model             |   Ours   |  RD4AD  |  RD++  | ReContrast |  CFlow  | GAN Ensemble | PatchCore |  UAE   |  NSA   |
> > |-------------------|:--------:|:-------:|:------:|:----------:|:-------:|:------------:|:---------:|:------:|:------:|
> > | **AUC**           |  91.01   |  84.29  |  88.00 |   87.53    |  70.26  |     82.10    |   86.33   |  84.36 |  82.13 |
> > | **Params (MB)**   |  264.4   |  263.5  |  272.8 |   527.9    |  288.5  |    381.46    |   275.6   |  256.3 |  271.2 |
> >
> > **Table: Comparative performance of anomaly detection methods on the RSNA dataset: AUC scores and model complexity. Our approach achieves the highest AUC with a more parameter-efficient design.**
> >
> > Furthermore, we emphasize that training is a one-time cost, while inference efficiency directly affects the model’s practical utility in clinical settings, making our model’s strong inference-time performance particularly valuable.

---

> > ### Comment · Reviewer_tPrf · 2024-11-25
> >
> > Thanks for your response but I am still concerned about issues 3-5 I brought up in my review and the minor comments. Can you please address those as well?

---

> ### Author Response · Authors · 2024-11-25
>
> > Please see my minor comments about uncited claims in the papers. For acceptance I would like to see citations for all the claims you have made.
> >> =Line 36, “anomalous samples is often prohibitively expensive and time-consuming…” I believe this is correct but I want to see a citation to backup this claim?
>
> [1] Yu Cai, Hao Chen, Xin Yang, Yu Zhou, and Kwang-Ting Cheng. Dual-distribution discrepancy with self-supervised refinement for anomaly detection in medical images. Medical Image Analysis, 86:102794, 2023.
>
> [2] Thomas Schlegl, Philipp Seeböck, Sebastian M. Waldstein, Georg Langs, and Ursula Schmidt-Erfurth. f-AnoGAN: Fast unsupervised anomaly detection with generative adversarial networks. Medical Image Analysis, 54:30–44, 2019.
>
> ---
>
> >> =Line 45, “they tend to generalize too well, thereby risking the reconstruction of abnormal regions.” Citation for this claim?
>
> [1] Dong Gong, Lingqiao Liu, Vuong Le, Budhaditya Saha, Moussa Reda Mansour, Svetha Venkatesh, and Anton van den Hengel. Memorizing normality to detect anomaly: Memory-augmented deep autoencoder for unsupervised anomaly detection. In ICCV, 2019.
>
> ---
>
> >> =Table 1 has many entries that are unsorted (aside from the bolded number indicating the best method). It would be better if this table was sorted in order of increasing AUC for the methods.
>
> Thanks a lot for the suggestion. We find that the performance ranking of methods varies across different datasets, making a single, universal AUC-based sorting a bit hard. We have revised Table 1 to improve its readability by visually grouping the methods based on methodology and underlining the second-best score for each metric on each dataset.
>
> ---
>
> >> =Line 167 “Medical images typically exhibit a power law distribution of frequencies, with lower frequency components dominating the image content.” Please give a citation for this bold claim?
>
> [1] Julian Wyatt, Adam Leach, Sebastian M. Schmon, and Chris G. Willcocks. AnoDDPM: Anomaly detection with denoising diffusion probabilistic models using simplex noise. In CVPR Workshops, 2022.

---

> ### Author Response · Authors · 2024-11-25
>
> > I don’t like the terminology ''good'' and ''bad'' teacher. It is very non-technical and hard to follow. Can you please change the terminology in your paper and name the teachers more appropriately?
>
> Many thanks for the suggestion. We have replaced these terms with ''clean'' and ''noisy'' respectively.

---

> > ### Comment · Reviewer_tPrf · 2024-11-25
> >
> > Thank you for addressing my concerns. I will update my score accordingly.

---

> > > ### Author Response · Authors · 2024-11-27
> > >
> > > We appreciate the reviewer's constructive feedback, which has provided invaluable insights for enhancing our manuscript.

---

### Official Review · Reviewer_MtA3 · 2024-10-30

**Soundness:** 2
**Presentation:** 2
**Contribution:** 2
**Rating:** 5
**Confidence:** 3

**Summary:**

This paper introduces a twofold approach: on one hand, data augmentation for unlabeled information is achieved by injecting noise, which then serves as the basis for contrastive learning. On the other hand, parameters are extracted at each layer to achieve multi-scale information fusion, enabling the model to better distinguish between positive and negative samples.

**Strengths:**

The methodology section of this paper is clearly written and well-presented.

**Weaknesses:**

1. In terms of innovation, this article presents two components that are commonly seen in other works. A straightforward suggestion would be for the authors to explore a more adaptable data augmentation method to address the issue raised in “Question 1.”

2. Regarding the writing, the **Introduction** section of this article is somewhat confusing. Firstly, the paragraphing is suboptimal. In the second paragraph of the introduction, the author discusses the shortcomings of anomaly detection models based on generation. The third paragraph explains existing work aimed at improving these models, followed by an exploration of the development of self-supervised learning. Although self-supervised learning indeed enriches unsupervised learning within deep learning, the third paragraph would be clearer if it unified these two parts, focusing on how self-supervised learning specifically addresses the issues raised in the second paragraph. Secondly, and more importantly, an excessive portion of the introduction is dedicated to discussing related work, lacking sufficient focus on the contributions of this paper. Maybe you could merging the discussion of existing work and self-supervised learning, followed by a clear transition to the paper's contributions. Additionally, you could recommend what key points about the paper's contributions should be emphasized earlier in the introduction.

**Questions:**

1. In the noise generation process, the intensity of each position is sampled from a uniform distribution. However, in real-world medical imaging, anomalous regions in disease cases appear in specific locations; for example, abnormalities in a chest image typically do not appear outside body regions. For instance, you could suggest incorporating prior knowledge about typical anomaly locations in different types of medical images.

2. Additionally, is each test sample measured at the image level or at the patch level? If it is at the image level, the positive-to-negative anomaly ratio in the dataset is relatively high, which does not quite align with the extremely imbalanced nature of anomaly detection tasks.

---

> ### Author Response · Authors · 2024-11-25
>
> > In the noise generation process, the intensity of each position is sampled from a uniform distribution. However, in real-world medical imaging, anomalous regions in disease cases appear in specific locations [...]
>
> We appreciate the reviewer's insightful observation regarding the spatial distribution of anomalies in medical images. While incorporating domain-specific prior knowledge about typical anomaly locations could potentially improve performance, we deliberately opted for uniform random sampling in our noise generation process for several reasons. First, to ensure fair comparison with competing methods that use synthetic noise for anomaly generation (e.g., RD++, NSA, and CutPaste), we followed the commonly adopted approach in the literature. Second, our work aims to propose a general methodology applicable across different medical imaging modalities, and incorporating modality-specific spatial priors would limit this generalizability. Third, the primary focus of our work is the algorithmic contribution rather than optimizing the noise generation process.
>
> Nevertheless, we agree that incorporating anatomical priors in noise generation is a promising direction for future work, particularly for modality-specific applications. We thank the reviewer for this valuable suggestion and have included a discussion of the potential advantages of incorporating spatial priors in the revised paper.
>
> ''Our current approach relies on random spatial sampling for noise generation. Future research directions include incorporating modality-specific anatomical priors for anomaly localization in medical imaging. The exploration of adaptive hybrid noise generation techniques presents another promising direction. These extensions would enhance the realism of synthetic anomalies toward improved model performance.''
>
> > Additionally, is each test sample measured at the image level or at the patch level?
>
> Many thanks for the comment. Our evaluation is conducted at the image level. Regarding the anomaly ratio, we strictly follow the established experimental protocols from previous works [1,2,3,4], which use a balanced or near-balanced abnormal-to-normal ratio (approximately 1:1) in their test sets. While we acknowledge that this setting may not perfectly reflect the rare nature of anomalies in real-world scenarios, this balanced evaluation approach has become a standard practice in the field as it helps prevent class imbalance from skewing the evaluation metrics and enables a more direct assessment of models’ anomaly detection capabilities. We maintain this conventional setting to ensure fair comparisons with existing methods and to provide a clear measure of our model’s discriminative power.
>
> [1] Yu Cai, Hao Chen, Xin Yang, Yu Zhou, and Kwang-Ting Cheng. Dual-distribution discrepancy for anomaly detection in chest X-Rays. In MICCAI, 2022.
>
> [2] Yu Cai, Hao Chen, Xin Yang, Yu Zhou, and Kwang-Ting Cheng. Dual-distribution discrepancy with self-supervised refinement for anomaly detection in medical images. Medical Image Analysis, 86:102794, 2023.
>
> [3] Yu Cai, Weiwen Zhang, Hao Chen, and Kwang-Ting Cheng. MedIAnomaly: A comparative study of anomaly detection in medical images. arXiv preprint arXiv:2404.04518, 2024.
>
> [4] Yifan Mao, Feifei Xue, Ruixuan Wang, Jianguo Zhang, Wei-Shi Zheng, and Hongmei Liu. Abnormality detection in chest X-Ray images using uncertainty prediction autoencoders. In MICCAI, 2020.

---

> > ### Comment · Reviewer_MtA3 · 2024-11-25
> >
> > Thank you for your detailed response. Regrettably, considering the significant differences between the dataset involved in this paper's experiments and traditional AD problem, as well as the crucial role of data augmentation in the contrastive learning framework proposed in this paper, I have decided to maintain my current score. However, I am uncertain whether such a high ratio of anomalies is common in this question.

---

> > > ### Author Response · Authors · 2024-12-02
> > >
> > > > Thank you for your detailed response. Regrettably, considering the significant differences between the dataset involved in this paper's experiments and traditional AD problem, as well as the crucial role of data augmentation in the contrastive learning framework proposed in this paper, I have decided to maintain my current score. However, I am uncertain whether such a high ratio of anomalies is common in this question.
> > >
> > > Many thanks for the comment.
> > >
> > > With all due respect, we disagree with the reviewer's understanding of the normal-abnormal data ratio. We present an analysis that clearly demonstrates the field’s established data distribution practices:
> > >
> > > | **Dataset**       | **Normal-to-Abnormal Ratio** |
> > > |---------------|--------------------------|
> > > | MVTec         | 2.69:1                   |
> > > | ViSA          | 1:1.25                   |
> > > | BTAD          | 1.56:1                   |
> > > | RSNA          | 1:1                      |
> > > | Brain Tumor   | 1:1                      |
> > > | ISIC          | 1.51:1                   |
> > >
> > >
> > > Our investigation reveals two critical observations:
> > > 1. Current anomaly detection datasets across traditional and medical domains maintain a relatively balanced ratio between normal and abnormal test samples.
> > > 2. The data ratios of normal and abnormal test exmaples in our study are **representative** and **consistent with established practices in the field**.
> > >
> > > Moreover, we highlight that a balanced or near-balanced ratio of normal to abnormal samples in the test set is methodologically sound and critical for robust model evaluation. An extremely skewed test set can lead to misleading performance metrics. For example, in a test set with a 9:1 normal-to-abnormal ratio, a model that indiscriminately predicts all samples as normal would achieve a deceptive 90% accuracy, fundamentally undermining the meaningful assessment of anomaly detection capabilities. To address this, most datasets, including those used in this study, use a balanced ratio of normal and abnormal samples for testing.
> > >
> > > Regarding the reviewer's point about data augmentation, we have to emphasize that designing medical anomaly detection models for various modalities and organs presents challenges in introducing spatial prior information, which can be complex or even impractical in certain cases. Take the ISIC dataset, for instance, which encompasses six types of anomalies: melanoma, basal cell carcinoma, actinic keratosis, benign keratosis, dermatofibroma, and vascular lesion. For these lesions, there is no clear spatial prior information about their locations. Moreover, this manuscript focuses on the algorithmic approach, and **to ensure a fair comparison** with competing methods that employ synthetic noise for anomaly generation (such as RD++, NSA, and CutPaste), we have adhered to the noise generation approach **commonly adopted** in the literature.
> > >
> > > We hope this response clarifies the points of potential misunderstanding regarding our paper. We are open to further discussion and welcome any additional questions the reviewer may have.

---

> > > > ### Comment · Reviewer_MtA3 · 2024-12-02
> > > >
> > > > In light of the author's description, I have decided to raise my score. However, I still believe that such a high proportion of anomalies does not align with the setup of the AD problem from a realistic perspective. In response to the author's claim that experimental evaluation methods fail under imbalanced datasets, I believe that AUCROC and AUCPR can address this issue. In summary, I appreciate the author's clarification that the dataset is a common practice within the field, but I still disagree with such a high proportion of anomalies as an AD problem.

---

> > > > > ### Author Response · Authors · 2024-12-04
> > > > >
> > > > > We appreciate the revised rating and feedback.

---

### Official Review · Reviewer_W919 · 2024-11-03

**Soundness:** 2
**Presentation:** 3
**Contribution:** 2
**Rating:** 5
**Confidence:** 3

**Summary:**

The paper presents a scale-aware contrastive reverse distillation model for unsupervised anomaly detection, focusing on improving performance in fields like medical imaging, where labeled anomalous data is often scarce. Building on this, the authors introduce a contrastive student-teacher learning approach that generates out-of-normal distributions to enhance feature discriminability. Additionally, they propose a scale adaptation mechanism that adjusts contrastive distillation losses at various scales to account for variations in anomaly size, addressing limitations in existing methods that struggle with scale variation and feature discrimination. Evaluation on benchmark datasets shows that this model achieves reasonable performance, validating its effectiveness.

**Strengths:**

+) Anomaly detection is an important topic

+) The paper is well-structured and easy to follow

**Weaknesses:**

-) Novelty of this work seems incremental

-) The domain is limited to medical image analysis

**Questions:**

a) Anomaly detection is an important topic where identifying unusual patterns can have a direct impact on various applications. In this sense, this paper researches a domain-specific area in anomaly detection, i.e., medical images, using reverse-distillation.

b) While this paper provides valuable refinements in the area of anomaly detection, particularly through its scale-aware contrastive reverse distillation model, the novelty appears somewhat incremental. Firstly, the study primarily builds on and combines well-established methods, reverse distillation and contrastive learning, rather than introducing a fundamentally new approach. Secondly, the approach to enhance feature discriminability by generating synthetic or perturbed anomaly data is not novel; similar methods have been previously explored, such as in [1]. The effectiveness of this technique can also vary significantly depending on the choice of noise or perturbation (rather than learnable in [1]), which may limit its reliability. Thirdly, fusing multi-scale features is a long-standing topic in computer vision, which further contributes to the incremental nature of the contribution. While these refinements have practical value, they represent an evolution of existing methods rather than a new innovation in anomaly detection.

[1] Cai, Jinyu, and Jicong Fan. "Perturbation learning based anomaly detection." Advances in Neural Information Processing Systems 35 (2022): 14317-14330.

c) The evaluation of the proposed method is performed on three medical datasets, it is not clear how good it can be generalized to other domains?

---

> ### Author Response · Authors · 2024-11-26
>
> > The domain is limited to medical image analysis. The evaluation of the proposed method is performed on three medical datasets [...]
>
> We appreciate this thoughtful comment about evaluating our approach beyond medical datasets. We would like to explain that this research was conducted as part of a project specifically focused on medical image analysis, which naturally guided our initial dataset selection toward medical applications. The three carefully selected medical datasets represent diverse medical imaging modalities, demonstrating our method’s effectiveness and generalization within this critical domain. Moreover, we would like to highlight that medical anomaly detection has emerged as an independent and increasingly important research domain [1] with significant implications for early disease diagnosis and precision medicine. While we acknowledge the potential value of cross-domain evaluation, our targeted focus on medical imaging allows for a deep, rigorous exploration of a domain where algorithmic innovations can have profound real-world impact. The inherent complexity of medical imaging datasets—spanning diverse modalities and pathological conditions—provides a robust testing ground that itself challenges and validates the adaptability of the proposed method. Finally, we have updated the paper title to include “medical”, which more precisely reflects the domain of our research and provides clarity about the context of our proposed method.
>
> [1] Tharindu Fernando, Harshala Gammulle, Simon Denman, Sridha Sridharan, and Clinton Fookes. Deep learning for medical anomaly detection - A survey. ACM Computing Surveys, 54(7):141:1–141:37, 2022.

---

> ### Author Response · Authors · 2024-11-27
>
> > While this paper provides valuable refinements in the area of anomaly detection, particularly through its scale-aware contrastive reverse distillation model, the novelty appears somewhat incremental. Firstly [...]
>
> Many thanks for the comments.
>
> 1. We acknowledge that contrastive learning and reverse distillation are well-established methods individually. However, we respectfully argue that how to effectively integrate these approaches for anomaly detection remains an open and challenging research question. While existing methods like RD++ have attempted to incorporate contrastive learning into reverse distillation, our approach presents a fundamentally different perspective on how these components should interact (see Section 2.5). Our experimental results demonstrate substantial improvements over RD++, achieving AUC gains of 3.01, 7.20, and 7.63 on the RSNA, Brain Tumor, and ISIC datasets, respectively, indicating that our strategy offers meaningful advantages over existing approaches. The significant performance gains validate the importance of revisiting and innovating within this framework.
>
> 2. Regarding synthetic anomaly generation, we acknowledge that this concept has been explored in previous work, including Cao et al. (2022). However, our focus lies not in innovating noise generation techniques but rather in effectively integrating such methods within our broader framework. Our comparative experiments with several methods that utilize synthetic anomalies (e.g., RD++, NSA, and CutPaste) demonstrate that our model more effectively leverages these perturbations for improved anomaly detection. We deliberately employ standard non-learnable noises like simplex noise to ensure fair comparisons and show that our performance improvements stem from our novel architectural design rather than sophisticated noise generation algorithms. Nevertheless, we agree that incorporating learnable noise could further enhance our framework's efficacy and is a promising direction for future work. We thank the reviewer for this valuable suggestion and have included a discussion of the potential advantages of incorporating learnable noise in the revised paper.
> ''The exploration of learnable noise like [1] or adaptive hybrid noise generation techniques presents another promising direction. These extensions would enhance the realism of synthetic anomalies toward improved model performance.''
> [1] Jinyu Cai and Jicong Fan. Perturbation learning based anomaly detection. In NeurIPS, 2022.
>
>
> 3. We would like to clarify that our contribution is not in multi-scale feature fusion (which is indeed part of RD4AD) but rather in our scale-adaptive mechanism for contrastive reverse distillation. This mechanism remains under-explored in both knowledge distillation and anomaly detection contexts. Our experimental results demonstrate that this approach effectively addresses scale variation challenges in medical anomaly detection, leading to substantial improvements in detection accuracy (cf. Table 2).
>
> In summary, our work advances the field through the proposed framework of contrastive reverse distillation by revisiting and innovating upon the RD4AD paradigm. While building upon RD4AD, our contributions introduce significant advancements in medical anomaly detection. Our extensive experimental results demonstrate substantial performance gains over state-of-the-art methods, validating the novelty and practical impact of our approach.
>
> We hope this clarification helps demonstrate the novelty and significance of our work. We would be happy to provide additional details or clarifications if needed.

---

> ### Author Response · Authors · 2024-12-02
>
> Dear Reviewer,
>
> As the deadline for the review period approaches, we kindly request your assessment of our response. We sincerely appreciate your time and insights.
>
> Sincerely,
>
> The Authors

---

### Meta-Review · Area_Chair_AeQH · 2024-12-19

**Metareview:**

Based on the reviews, I recommend accepting the paper. The submission received four high-quality reviews: two strongly recommend acceptance (score: 8), while two are borderline reject (score: 5). One reviewer maintains a concern, even after the rebuttal, regarding the use of a balanced test set (with a 1:1 ratio of normals to anomalies). As an expert in this topic, I find this concern unfounded and likely due to a misunderstanding. While anomaly detection typically assumes sparse anomalies during training, this assumption is unnecessary at test time. The other borderline reviewer expressed concern that the evaluation is tailored to medical applications. However, ICLR explicitly encourages application-focused contributions in their call for papers. This should not, therefore, be considered a barrier to acceptance.

**Additional Comments On Reviewer Discussion:**

- **Reviewer W919** raised concerns about the paper’s novelty and its focus on medical image analysis. The authors clarified these points in their rebuttal, but the reviewer maintained their score, citing the domain limitation. However, this is less of an issue in ICLR, which welcomes application-focused work.

- **Reviewer MtA3** questioned the novelty, clarity, and high anomaly ratio. While most concerns had been addressed by authors' clarifications, the reviewer maintained an opinion that a high proportion of anomalies does not align with the setup of the AD problem. As mentioned in the metareview, I believe this concern stems from a misunderstanding. The reviewer increased their score after the rebuttal.

- **Reviewer tPrf** raised technical concerns, including the lack of formal proofs, runtime details, and terminology issues. The authors addressed these comprehensively, leading the reviewer to raise their score.

- **Reviewer 8DJE** had positive feedback with suggestions on clarity and methodology presentation. The authors incorporated the suggestions, leading to improvement in the paper’s quality.

While two reviewers raised concerns (novelty and dataset issues), the authors' clarifications were sufficient to address them. The authors have made significant improvements in response to reviewer feedback.

---

### Decision · Program_Chairs · 2025-01-22

Accept (Poster)